# Robust Contrastive Learning Using Negative Samples with Diminished Semantics

**Songwei Ge**
Univeristy of Maryland
songweig@cs.umd.edu

**Shlok Mishra**
Univeristy of Maryland
shlokm@cs.umd.edu

**Haohan Wang**
Carnegie Mellon University
haohanw@cs.cmu.edu

**Chun-Liang Li**
Google Cloud AI
chunliang@google.com

**David Jacobs**
Univeristy of Maryland
dwj@cs.umd.edu

## Abstract

Unsupervised learning has recently made exceptional progress because of the development of more effective contrastive learning methods. However, CNNs are prone to depend on low-level features that humans deem non-semantic. This dependency has been conjectured to induce a lack of robustness to image perturbations or domain shift. In this paper, we show that by generating carefully designed negative samples, contrastive learning can learn more robust representations with less dependence on such features. Contrastive learning utilizes positive pairs that preserve semantic information while perturbing superficial features in the training images. Similarly, we propose to generate negative samples in a reversed way, where only the superfluous instead of the semantic features are preserved. We develop two methods, texture-based and patch-based augmentations, to generate negative samples. These samples achieve better generalization, especially under out-of-domain settings. We also analyze our method and the generated texture-based samples, showing that texture features are indispensable in classifying particular ImageNet classes and especially finer classes. We also show that model bias favors texture and shape features differently under different test settings. Our code, trained models, and ImageNet-Texture dataset can be found at https://github.com/SongweiGe/Contrastive-Learning-with-Non-Semantic-Negatives.

## 1 Introduction

Recent studies on self-supervised learning have shown great success in learning visual representations without human annotations. The gap between unsupervised and supervised learning has been progressively closed by contrastive learning [53, 48, 7, 18, 49, 6, 16, 56]. In the meantime, CNNs trained in the supervised setting are known to learn correlations between labels and superfluous features such as local patches [4, 3], texture [14], high-frequency components [51], and even artificially added features [26], which has raised concerns about deploying these models in a real scenario [30, 15]. CNNs trained by contrastive learning methods are no exception [23]. In this paper, we propose to construct negative samples that only preserve non-semantic features. We show that using contrastive learning methods trained with these negative samples can mitigate these concerns.

Contrastive learning methods exploit carefully designed augmentations to construct positive pairs and pull their representations together. These augmentations are crucial to contrastive learning [7, 6]. A common assumption behind these augmentations is to preserve the semantics of the input images while perturbing other superficial signals. This inspires us to generate negative samples and inject additional implicit biases on the visual features learned by the models. Specifically, we utilize

35th Conference on Neural Information Processing Systems (NeurIPS 2021).

Semantic positive samples       Non-semantic negative samples

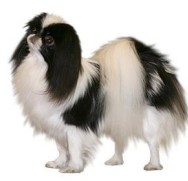 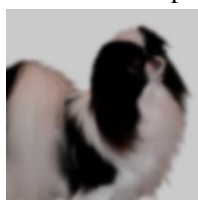 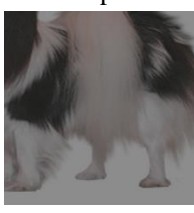 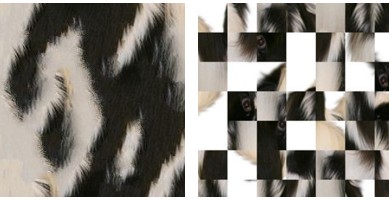

(a) Input image     (b) Query sample     (c) Positive sample     (d) Texture-based NS (e) Patch-based NS

Figure 1: We propose to construct negative samples (NS) from input images for contrastive learning with augmentations that only preserve non-semantic information such as texture and local features.

augmentations that diminish the semantic features while keeping the undesired features such as texture. By pushing apart the representations of such negative samples and input images, the models are expected to rely more on the semantics of the images and less on superficial features.

Inspired by the non-semantic features, we propose two methods to craft negative samples. The first method relies on texture synthesis tools from classic approaches [12, 52]. It generates realistic texture images based on two patches extracted from input images, as shown in Figure 1(d). For each image in ImageNet, we generate its texture version and form a dataset which we call ImageNet-Texture. The second method constructs non-semantic images by tiling randomly sampled patches of different sizes from the input image, as shown in Figure 1(e). Comparing the non-semantic negative samples with two semantic positive samples in Figure 1(b) and Figure 1(c), the dog from the input image is still recognizable in the positive samples but hard to understand from negative samples. Instead, local statistics such as the fur and color of the dog are preserved in the negative samples.

The generated non-semantic samples can be readily used with existing contrastive learning methods that distinguish positive pairs from negative pairs such as MoCo [18] and SimCLR [7]. Despite their simplicity, we show that these non-semantic negative samples are actually harder than the standard negative samples used by these methods, which are inefficient at leveraging hard negatives [13, 31]. Further, our negative pairs can also be used by contrastive learning methods that do not explicitly use negative samples, such as BYOL [16]. We evaluate our methods with two contrastive learning methods, MoCo [18, 8] and BYOL [16], on three datasets, ImageNet-100, ImageNet-1K and STL-10. When using our proposed augmentations to generate negative samples and minimize their representation similarity to the input images, we notice a consistent improvement on the generalization performance over backbone methods [8, 16] and previous negative example generation strategies [31, 43], especially under out-of-distribution (OOD) settings.

We conduct a systematic analysis of how the shape-texture trade-off influences model performance based on the proposed ImageNet-Texture dataset. We control the penalty on similarities between the non-semantic negative examples and the query samples. This impacts the trade-off between using shape and texture features. We find that the relative importance of texture and shape features varies across different datasets. For example, shape bias benefits ImageNet-Sketch [50] more than the original ImageNet validation set. On the other hand, texture bias benefits finer-grained classification more, such as dog breed classification included in ImageNet. These results complement previous evidence showing the effectiveness of shape features in classifying 16 coarse classes [14]. Such preference for one feature over the other is also observed intra-dataset: the texture is more important for some classes such as dishrag and plaque. These observations make us question the relationship between shape and texture features as the implicitly necessary bias of CNNs and advocate for an adaptive combination of both when deploying the model in real scenarios. In summary:

- We propose texture-based and patch-based augmentations to generate negative samples from input images, and show that these negative samples improve the generalization of contrastive learning.

- We introduce the ImageNet-Texture dataset, which contains texture versions of ImageNet images generated by texture synthesis tools.

- We provide fine-grained analysis on the shape-texture trade-off of CNNs, and show different scenarios when one is preferred over the other.

## 2  Negative Samples with Diminished Semantics

CNNs are apt to learn low level features such as texture under supervised settings [3, 14, 51]; this has been recently witnessed under the contrastive learning setting as well [23]. To mitigate this problem, we propose two methods, texture-based and patch-based augmentations, to generate negative samples for contrastive learning. Texture-based augmentation generates realistic images based on texture synthesis and patch-based augmentation exploits more comprehensive local features by sampling patches from input images. By penalizing learned similarities between the representations of images and their non-semantic counterparts, the model is encouraged to rely less on the undesired features and focus more on the semantics. In practice, we find the two negative samples play similar roles and the patch-based method works slightly better. In this section, we start with an overview of contrastive learning and show how non-semantic negatives are used in these frameworks. Then we elaborate on the two approaches to generate negative samples with diminished semantics.

### 2.1  Contrastive learning with non-semantic negatives

Given an encoder network $f$ and an image $\mathbf{x}$, we denote the output of the network as $\mathbf{z} = f(\mathbf{x})$. We use $z_i$ and $z_p$ to denote the representations of the query sample $x_i$ and a positive sample $x_p$ generated from the same input image with augmentations that preserve semantics. For contrastive learning methods like MoCo [18] and SimCLR [7], $z_n$ denotes the representation of the standard negative sample $x_n$ extracted from the memory bank (MoCo) or other images in the current batch (SimCLR). $z_{ns}$ is the representation of the proposed negative sample $x_{ns}$ which contains particular non-semantic features of the input image with the semantic part weakened. We extend the noise-contrastive estimation (NCE) loss as below:

$$\mathcal{L}_{\text{NCE}} = -\sum_{i \in I} \log \frac{\exp\left(z_i^T z_p/\tau\right)}{\exp\left(z_i^T z_p/\tau\right) + \exp\left(\alpha z_i^T z_{ns}/\tau\right) + \sum_{n \in \mathcal{N}} \exp\left(z_i^T z_n/\tau\right)}, \tag{1}$$

where $\tau$ is a temperature parameter and $\alpha$ is an additional scaling parameter for non-semantic negatives. A larger $\alpha$ implies a stronger penalty on the similarity between the representations of the query image and its non-semantic version. In Appendix B.1 we discuss other possible ways to apply $\alpha$.

Methods like BYOL [16] do not explicitly rely on negative samples. Nevertheless, BYOL adapts the loss to maximize the agreement of positive pairs. Therefore, we explicitly use the non-semantic negative sample with their loss to minimize its similarity to the query sample:

$$\mathcal{L}_{\text{BYOL}} = \|z_i - z_p\| - \alpha\|z_i - z_{ns}\| = 2 - 2\alpha - 2z_i^T z_p + 2\alpha z_i^T z_{ns}. \tag{2}$$

We overload $\alpha$ to be the parameter that controls the penalty on the similarity between the representations of input image and its non-semantic version under BYOL, with similar intention as MoCo and SimCLR above. To minimize either $\mathcal{L}_{\text{NCE}}$ or $\mathcal{L}_{\text{BYOL}}$, the encoder must learn features from $x_i$ that are not contained in $x_{ns}$ but shared with $x_p$.

### 2.2  Texture-based negative sample generation

We use texture synthesis tools to generate negative samples. Texture synthesis aims to generate realistic images that preserve as much local structure as possible from an example image [19, 11, 42]. For instance, as shown in Figure 1(d), the texture of the input dog image preserves the fur and colors of the dog. Notably, in previous discussion of robustness [3, 50], such local structure has often been recognized as highly correlated with the labels yet superfluous to generalization. For example, under large domain shift due to lighting, motion, and even modality, the texture is more apt to change than the semantic features, such as the shape. Furthermore, CNNs trained on ImageNet are more likely to classify images based on the texture features rather than the shape features which are instead preferred by humans due to their transferability [14]. To encourage the model to rely more on the shape features, we propose a two-step method to generate the texture image of input images as the negative samples for contrastive learning.

To be specific, we first sample two patches from given images as the input to the texture synthesis algorithms. One patch is extracted from the center of the image. This patch is expected to reflect the texture of the object according to the implicit bias contained in the ImageNet dataset that most

of the objects are center-oriented in the images [2]. The other patch is extracted from a random location to reflect other possible textures of the image (e.g. background, peripheral region of the object). In this work, we extract patches with size $96 \times 96$ when image size allows, otherwise $48 \times 48$ patches are extracted. Second, we adopt off-the-shelf texture synthesis algorithms [12, 52, 1] to generate texture images based on the two patches. These non-parametric algorithms iteratively sample pixels from given patches that share a similar neighborhood with the current pixel. Specifically, we use the open-source software built on these methods [12, 52, 1] with multi-threaded CPU support implemented in Rust [1]. For each sample in the ImageNet dataset, we generate one $224 \times 224$ texture image to construct a dataset that has the same training and validation size as the ImageNet dataset. We call this dataset ImageNet-Texture. More examples can be found in Appendix A.1.

### 2.3 Patch-based negative sample generation

To simulate the local information contained in the images [4, 3], we propose an efficient patch-based method to generate non-semantic images. Given an image and a patch size $d$, we sample patches of size $d$ from $(\lceil \frac{224}{d} \rceil)^2$ non-overlapping random locations that lie entirely in the image. The patches are then tiled and cropped into $224 \times 224$ as negative samples. Compared with the texture-based method, this generation process takes negligible time, therefore it can be implemented as part of the data loading process in parallel with training. By doing so, each training sample can be paired with different negative samples generated from different patches every time it is used, compared with the two fixed patches selected when generating texture images.

Different from the texture-based method that generates realistic images, the patch-based method generates images with artificial lines as shown in Figure 1e. One might be concerned with possible degenerate solution where the model outputs a low similarity whenever it detects the repeated sharp changes in the horizontal or vertical directions, which could be done with a single layer of convolution. However, interestingly, we find that the model does not find such a simple solution in practice. This is also noticed in a previous study where the image and its copy with a patch cut out are non-trivially distinguished by the model [34]. To mitigate this potential issue, we randomly sample patch size $d$ from a prior distribution instead of using a fixed $d$ in practice, which allows the model to look at texture at different scales.

### 2.4 How hard are the texture-based and patch-based negative samples?

Constrastive learning methods are known to struggle with finding hard negative samples [13, 31] and researchers have proposed several ways to better leverage hard negatives [31, 43]. An intermediate question is how hard are our proposed negative samples compared with those standard negative samples used in previous constrastive learning methods [7, 18], i.e. random training samples.

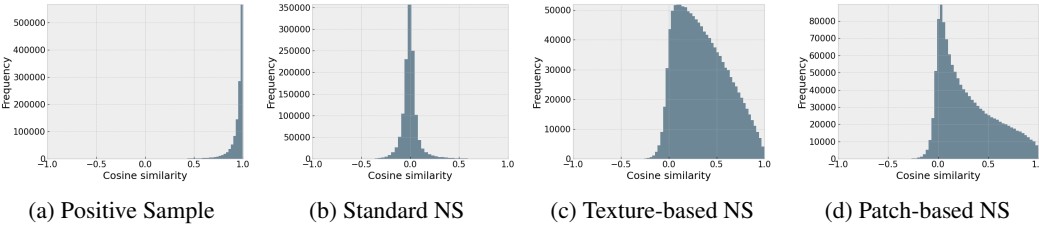

(a) Positive Sample    (b) Standard NS    (c) Texture-based NS    (d) Patch-based NS

Figure 2: The histogram of cosine similarity between the representations of query sample and its paired samples, namely positive sample and standard, texture-based, and patch-based negative samples (NS), using MoCo-v2 model trained on the ImageNet-1K dataset for 200 epochs.

We use the official MoCo-v2 model pretrained on the ImageNet-1K dataset for 200 epochs to calculate the cosine similarities between different kinds of pairs across the ImageNet training set. We plot the histogram of these similarities in Figure 2. As shown in the Figures 2a and 2b, most positive pairs and negative pairs have similarity close to 1 and 0 respectively. Specifically, the average similarities across the training samples are $0.94257$ and $0.00018$ for positive and negative pairs. As shown in

---

[1]https://github.com/EmbarkStudios/texture-synthesis

the Figures 2d and 2c, the distributions of patch-based and texture-based negative samples are very different from those of standard negative samples; their similarity distributions have heavy tails in the positive region. Specifically, the distribution of patch-based and texture-based negative samples have average similarity 0.29503 and 0.35248 across the dataset, which shows that they remain difficult after training with standard negative examples.

## 3 Experiments

In this section, we evaluate the two kinds of non-semantic negative samples with two contrastive learning methods, MoCo and BYOL, on the ImageNet dataset. We also experiment using its subset, the ImageNet-100 dataset [48, 31], which allows us to perform more comprehensive experiments. We report accuracy on out-of-domain (OOD) datasets including the ImageNet-C(orruption) [22], ImageNet-S(ketch) [50], Stylized-ImageNet [14], and ImageNet-R(endition)[21] datasets as an evaluation of the model's robustness to domain shifts. ImageNet-C and Stylized-ImageNet contain images transformed from the images in the ImageNet validation set with common corruption and transferred style. ImageNet-S and ImageNet-R are collected independently from the ImageNet dataset and share all or a subset of the classes in the ImageNet dataset with a focus on sketch and other rendering modalities. We show that with our proposed non-semantic negatives, contrastive learning generalizes better under domain shifts. For patch-based negatives, it also improves the performance on the in-domain dataset.

### 3.1 ImageNet-100

| | ImageNet | ImageNet-C | ImageNet-S | Stylized-ImageNet | ImageNet-R |
|---|---|---|---|---|---|
| MoCo-v2 - $k = 16384$ | $77.88_{\pm 0.28}$ | $43.08_{\pm 0.27}$ | $28.24_{\pm 0.58}$ | $16.20_{\pm 0.55}$ | $32.92_{\pm 0.12}$ |
| + Texture-based - $\alpha = 2$ | $77.76_{\pm 0.17}$ | $43.58_{\pm 0.33}$ | $29.11_{\pm 0.39}$ | $16.59_{\pm 0.17}$ | $33.36_{\pm 0.15}$ |
| + Patch-based - $\alpha = 2$ | $\mathbf{79.35}_{\pm 0.12}$ | $\mathbf{45.13}_{\pm 0.35}$ | $31.76_{\pm 0.88}$ | $17.37_{\pm 0.19}$ | $34.78_{\pm 0.15}$ |
| + Patch-based - $\alpha = 3$ | $75.58_{\pm 0.52}$ | $44.45_{\pm 0.15}$ | $\mathbf{34.03}_{\pm 0.58}$ | $\mathbf{18.60}_{\pm 0.26}$ | $\mathbf{36.89}_{\pm 0.11}$ |
| MoCo-v2 - $k = 8192$ | $77.73_{\pm 0.38}$ | $43.22_{\pm 0.39}$ | $28.45_{\pm 0.36}$ | $16.83_{\pm 0.12}$ | $33.19_{\pm 0.44}$ |
| + Patch-based - $\alpha = 2$ | $\mathbf{79.54}_{\pm 0.32}$ | $\mathbf{45.48}_{\pm 0.20}$ | $\mathbf{33.36}_{\pm 0.45}$ | $\mathbf{17.81}_{\pm 0.32}$ | $\mathbf{36.31}_{\pm 0.37}$ |
| MoCo-v2* | $80.00_{\pm 0.14}$ | $45.15_{\pm 0.42}$ | $30.38_{\pm 0.30}$ | $16.68_{\pm 0.39}$ | $30.38_{\pm 0.30}$ |
| + IFM [44] - $\epsilon = 0.05$ | $80.86$ | $47.36$ | $31.35$ | $18.18$ | $36.79$ |
| + IFM [44] - $\epsilon = 0.1$ | $81.22$ | $47.46$ | $31.87$ | $18.42$ | $37.23$ |
| + IFM [44] - $\epsilon = 0.2$ | $81.02$ | $47.19$ | $31.55$ | $\mathbf{18.68}$ | $37.14$ |
| + Patch-based - $\alpha = 2$ | $\mathbf{81.49}_{\pm 0.11}$ | $\mathbf{47.48}_{\pm 0.20}$ | $\mathbf{34.20}_{\pm 0.40}$ | $17.95_{\pm 0.41}$ | $\mathbf{38.45}_{\pm 0.19}$ |
| BYOL | $78.76_{\pm 0.28}$ | $44.43_{\pm 0.35}$ | $35.84_{\pm 0.38}$ | $15.01_{\pm 0.19}$ | $39.53_{\pm 0.51}$ |
| + Patch-based - $\alpha = 0.05$ | $\mathbf{78.81}_{\pm 0.33}$ | $\mathbf{44.60}_{\pm 0.21}$ | $\mathbf{36.76}_{\pm 0.51}$ | $\mathbf{15.52}_{\pm 0.22}$ | $\mathbf{41.16}_{\pm 0.39}$ |
| InsDis [53] | $68.52$ | $28.93$ | $16.67$ | $9.86$ | $19.60$ |
| CMC [48] | $79.34$ | $39.28$ | $24.04$ | $13.88$ | $32.68$ |
| InfoMin [49] | $82.74$ | $48.87$ | $38.43$ | $18.14$ | $40.68$ |
| Supervised | $86.26$ | $49.17$ | $34.95$ | $21.20$ | $39.76$ |

Table 1: Top-1 accuracy on the ImageNet-100 dataset and its OOD variants. We consider the supervised baseline as well as several self-supervised baselines including MoCo-v2, BYOL, InsDis, CMC, and InfoMIN. For our main comparison using MoCo-v2 and BYOL, we also report the standard deviation of 3 runs. For MoCo models, $k$ represents the size of the memory bank. We use * to denote the experiments that use the training setting in a concurrent work IFM [44].

We follow the hyperparameters used in [31] to train MoCo-v2 on the ImageNet-100 dataset with a memory bank size $k = 16384$ or a halved memory bank size. We also conduct experiments following the hyperparameters in a concurrent study [44] except that we keep $k = 16384$ for our method. For patch-based augmentation parameters, we use patch size sampled from a uniform distribution $d \sim \mathcal{U}(16, 72)$. The parameter $\alpha$ is indicated behind each model name. We discuss the impact of $\alpha$ in detail in the next section. More ablations on the patch-based augmentations can be found in Appendix C.3. For ImageNet-C, we report the average accuracy across 5 levels of corruption severity.

We repeat the experiments, including both the pretraining and linear evaluation, for 3 runs and report the mean and standard deviation in Table 1. As shown in the table, when following the

previous memory bank size, using both patch-based and texture-based negatives improve the OOD generalizations. Specifically, patch-based augmentation increases the accuracy on ImageNet-S by 5.79% and ImageNet-R by 5.97% when $\alpha = 3$. When $\alpha = 2$, it also increases the in-domain accuracy by 1.47% and accuracy on ImageNet-C by 2.05%. The similar trend shared by standard ImageNet and ImageNet-C with different $\alpha$ can be attributed to the resemblance of the images in the two dataset, especially those corrupted images with a lower level of severity. We show the performance of the model with $\alpha = 3$ is actually better on the highest corruption level as shown in Appendix C.2. The improvement achieved using texture-based negatives is less, probably because the information contained in the texture image is restricted due to the limited access to the two fixed patches. When the memory bank is halved to be $8,192$, the baseline MoCo model has slightly worse performance, decreasing from 77.88 to 77.73. But with patch-based hard negative samples, the MoCo-v2 model instead achieved the best accuracy 79.54 on the ImageNet-100 validation set, IamgeNet-C and IamgeNet-R. We discuss this more in a later section.

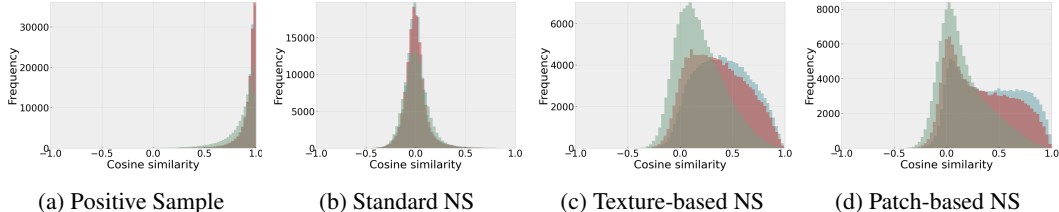

| (a) Positive Sample | (b) Standard NS | (c) Texture-based NS | (d) Patch-based NS |

Figure 3: Histogram of cosine similarity between the representations of query sample and its paired positive sample, standard, texture-based, and patch-based negative samples (NS), using models trained without (blue) and with patch-based negative samples (red: $\alpha = 2$, green: $\alpha = 3$).

Similar to Figure 2, in Figure 3 we visualize the distribution of the cosine similarity between the query sample and both semantic and non-semantic samples calculated based on the models trained with and without patch-based negative samples. The shift of the distribution towards the origin in Figure 3d meets the expectation that our method reduces the similarity of input images and patch-based negative samples. Specifically, the average similarity decreases from $0.4040$ to $0.3252$ to $0.1593$ when $\alpha$ increases from 0, namely no patched-based negatives, to 2 to 3. Interestingly, we notice in Figure 3c that the similarity to texture-based negative samples also decreases, and the average similarity decreases from $0.4114$ to $0.3541$ to $0.1896$, although we did not explicitly penalize it. This demonstrates the resemblance of patch-based negative samples to texture-based negative samples. Given the better performance achieved with patch-based negative samples, for the rest of the experiments, we mainly focus on the patch-based methods. But we still conduct our analysis on the texture-based samples. We also find a marginal decrease in the positive similarity $(-0.0068)$ and negative similarity $(-0.0008)$ when $\alpha = 2$ and a substantial decrease in the positive similarity $(-0.0737)$ and increase in the negative similarity $(0.0036)$ when $\alpha = 3$. A similar figure for texture-based negative samples can be found in the Appendix in Figure 12.

## 3.2 ImageNet-1K

|  | ImageNet | ImageNet-C | ImageNet-S | Stylized-ImageNet | ImageNet-R |
|---|---|---|---|---|---|
| MoCo-v2 [8] | 67.60 | 87.7 | 17.47 | 5.55 | 27.81 |
| + MoCHi [31] | 67.56 | 88.7 | 16.32 | 5.94 | 25.71 |
| + Patch-base NS - $\alpha = 2$ | **67.92** | **87.6** | **18.58** | **6.34** | **28.95** |

Table 2: Top-1 accuracy on the ImageNet-1K dataset and its sketch, stylized, rendition variants, and mCE on the ImageNet-C dataset.

We follow the official hyperparameters [8] to train MoCo-v2 with our patch-based negative samples on the ImageNet-1K dataset. For the parameter $\alpha$ and patch size $d$, we follow the same configuration used on the ImageNet-100 dataset. We compare our results against the MoCo-v2 baseline [8] and the hard negative mixing algorithm, MoCHi [31]. Due to limited computational resources, we report the metrics evaluated with the official model without repeated runs. The results are shown in Table 2.

More results can be found in Appendix C.6. Note that for ImageNet-C, we show the mCE metric [22], for which smaller is better. For the other datasets, we show the top-1 accuracy.

## 3.3 Extension to other non-semantic features

Non-semantic features are sometimes referred to as "shortcuts" in the contrastive learning literature [6, 7]. Models that leverage such features often exhibit unfavorable generalization to downstream tasks. For example, without color jittering, SimCLR [9] tends to utilize color histograms to reduce the training loss. In this section, we show that models trained with non-semantic negatives are coerced to avoid the shortcuts shared between query images and their non-semantic counterparts. In the example of the color shortcut, we

| Model | Top-1 Accuracy |
|---|---|
| MoCo-v2 [8] | 70.44 |
| + Patch-based NS | 76.42 |

Table 3: Test accuracy of MoCo-v2 on the ImageNet-100 dataset after removing color jittering and adding patch-based negatives.

note that the expected color distribution of our patch-based negatives is identical to that of the query images, and the actual distribution of samples is close. We conduct experiments with MoCo-v2 on the ImageNet-100 dataset while removing the color jittering from the augmentations. The accuracy of models with and without patch-based negatives are reported in Table 3. We found that patch-based negatives contribute significant effectiveness in preventing the models from learning such a color distribution shortcut.

## 3.4 Memory bank size

Contrastive learning methods based on negative samples suffer from ineffective excavation of hard negatives [13, 31] and resort to large batch sizes [7] or memory bank [18]. In this section, we study whether our proposed negative samples can mitigate this problem on the STL-10 and ImageNet-100 datasets. We keep the hyperparameters intact and vary the memory bank size. We report the accuracy of the MoCo-v2 baseline with and without patch-based negative samples on STL-10 dataset in Table 4. We also compare with [43] which exploits hard negatives through reweighting. We found that with proper hyperparameters the MoCo-v2 baseline already beats the reweighting results with SimCLR.

| | |
|---|---|
| SimCLR [7] | 80.16 |
| + Debiased [10] | 84.90§ |
| + Hard [43] | 87.42§ |
| MoCo-v2 [8] | 88.00 |
| + Patch-based NS | 89.36 |

Table 4: Top-1 accuracy on the STL-10 dataset.
§ denotes results visually extracted from Figure 2 in [43].

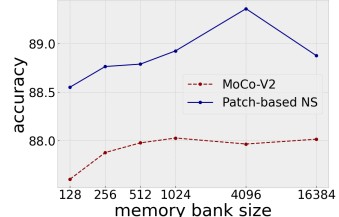

Figure 4: Top-1 accuracy on the STL-10 dataset with different memory bank sizes.

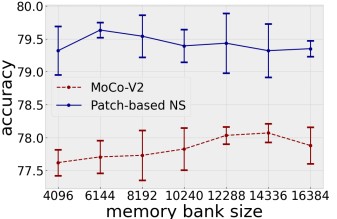

Figure 5: Top-1 accuracy on the ImageNet-100 dataset with different memory bank sizes.

As shown in Figures 4 and 5, using patch-based non-semantic negatives consistently improves the MoCo baseline when the number of standard negatives varies. When slightly decreasing the memory bank size on the STL-10 dataset as shown in Figure 4 and ImageNet-100 in Table 1, the performance with patch-based negatives increases. This is probably because, according to Eq. 1 and analysis in Appendix B.1, a smaller memory bank size causes a larger contribution of non-semantic negatives to the loss, and consequently a larger regularization. To further demystify this observation, we conduct experiments with evenly sampled memory bank sizes between 4096 and 16384 and report the average and standard deviation across 3 runs in Figure 5. We confirm a consistent recession of baseline accuracy when decreasing memory back sizes [18, 43]. However, the steady improvement led by the non-semantic negatives effectively mitigates the problem - the decrease caused by a smaller memory bank is less substantial and using patch-based negatives always beats the baseline.

# 4 Discussion

## 4.1 Controlling the shape-texture trade-off with $\alpha$

There has been a growing interest in understanding the cause and impact of the trade-off between shape and texture bias of CNNs [14, 23, 35, 27]. CNNs trained on ImageNet are known to be over-reliant on the texture features [14]. Contrastive learning with our non-semantic negatives serve as not only an effective method to reduce such reliance, but a natural tool to study such a trade-off.

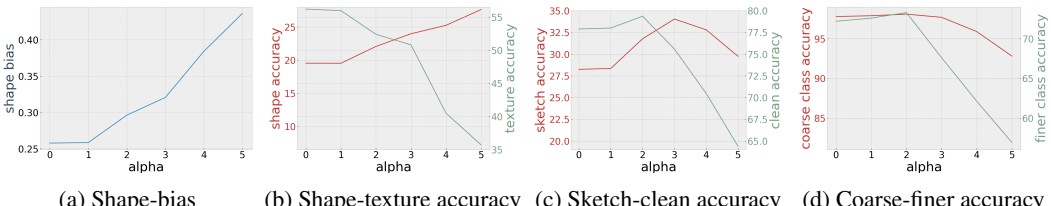

(a) Shape-bias    (b) Shape-texture accuracy  (c) Sketch-clean accuracy  (d) Coarse-finer accuracy

Figure 6: Larger $\alpha$ monotonically increases the model bias to shape features over texture features. Model performance is impacted by such a trade-off differently under different settings. In all scenarios, slightly calibrated shape bias improves model performance. The test settings represented in the red lines gain more from the increased shape bias than the settings represented in the green lines.

We train MoCo-v2 models with different $\alpha$ from 1 to 5 on the ImageNet-100 dataset. As shown in Figure 6a, we find that $\alpha$ effectively controls the trade-off on the model bias to the shape and texture features. $\alpha = 0$ is used to denote the baseline method. Specifically, a larger $\alpha$ in the loss function 1 leads to a larger penalty on the similarity between the representations of query samples and non-semantic samples, consequently a larger shape bias. We follow [14] to calculate shape bias on the stimuli images with conflicted shape and texture clues generated by style transfer. We show the corresponding accuracy on the shape and texture labels in Figure 6b.

As shown in Figure 6b, when $\alpha$ increases the texture accuracy on the stimuli dataset monotonically decreases while the shape accuracy monotonically increases. To further study how the trade-off between shape and texture bias impacts the model performance, we first compare the accuracy on the ImageNet validation dataset and ImageNet-Sketch dataset [50] when $\alpha$ varies in Figure 6c. We find that on both datasets, slightly increased shape bias over baseline ($\alpha = 2$) improves performances. Interestingly, the accuracy peak on the ImageNet-Sketch appears at $\alpha = 3$ while the peak appears at $\alpha = 2$ on the ImageNet validation dataset. In addition, for even larger $\alpha$ the accuracy on the ImageNet-Sketch dataset still outperforms the baseline while the standard accuracy gets hurt. This shows that different downstream tasks may benefit differently from differently shape-biased models.

We plot the histograms of similarities calculated by the models trained with different $\alpha$ in Appendix Figure 13. We find that large $\alpha$ makes the original pretext task challenging - the model cannot effectively pull together the representations of positive pairs. Specifically, when $\alpha$ increases from 1 to 5, the average similarity of the positive pairs decrease from 0.9267 to 0.7541. This demonstrates that it is hard for the model to learn representations that are completely independent of the texture features contained in the non-semantic images.

## 4.2 Rethinking the shape-texture trade-off through class-based analysis

The initial discussion on the shape-texture trade-off shows that humans rely more on the shape features while CNNs rely more on texture features and increasing shape bias can improve accuracy and robustness [14]. However, similar to [40, 23], we notice that increasing shape bias does not always improve the generalization and robustness of the models. To better understand this phenomenon, we provide two observations based on the analysis of the ImageNet-Texture dataset and our method to explain why a texture-biased model is helpful with classification on the ImageNet dataset.

First, we find that an increasing shape bias often leads to more errors among the fine-grained classes. The initial discussion of the shape-bias [14] only pays attention to the selected 16 coarse classes. We thus compare the finer and coarse class accuracy on the dog images of ImageNet dataset as in Figure 6d. For coarse class accuracy, the predictions are counted to be correct whenever the image is

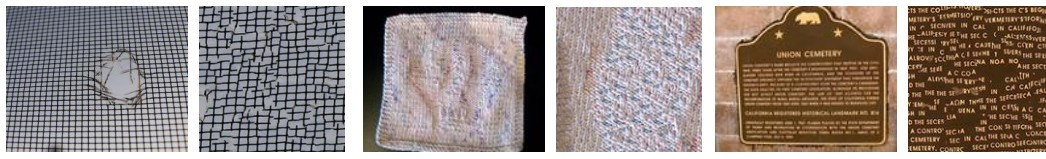

n04589890, window screen, 0.58    n03207743, dishrag, 0.82    n02892201, plaque, 0.64

Figure 7: A ResNet-50 model trained on the ImageNet dataset achieves decent accuracy when only texture features are available on some classes. A normal image and its texture version are displayed for some of these classes. The caption indicates the class ID, name, and accuracy on texture images.

classified as a dog class, no matter which dog class is predicted, while for the finer class accuracy, only those predictions of target dog classes are counted to be correct. We notice that the finer class accuracy drops more significantly when shape bias increases as opposed to the coarse class accuracy. For example, when $\alpha = 3$, the finer class accuracy drops from 72.9 to 67.6 while the coarse class accuracy slightly decreases from 97.8 to 97.7. Therefore, for datasets with numerous fine-grained classes like ImageNet, a texture-biased model is more helpful for a higher accuracy, which confirms the previous conjecture [54] . Second, in Appendix Figure 9, we show a scatter plot of texture accuracy vs. standard accuracy of different model architectures and a histogram of accuracy on individual ImageNet-Texture classes. We identify several classes where using only texture features is sufficient to achieve a high classification accuracy. These classes are all missing in the previous study [14]. As shown in Figure 7 and Appendix Figure 10, texture serves as a more important clue than the shape for these classes.

## 5    Related Work

**Contrastive learning based self-supervised learning**    Recent contrastive learning based self-supervised learning methods including MoCo [18], SimCLR [7], InfoMin[49], SimSiam[9], BYOL [16], SwAV [6], Barlow Twins [56] have proven helpful in learning visual representations. These methods rely on different pretext tasks to increase the agreement among the different views of the same image. The augmentations used to generate these views are essential to the success of these contrastive learning methods [7, 6] by preventing shortcuts such as the use of simple color histogram [7]. There is an ongoing trend of developing novel augmentations [6, 49] or adaptively applying augmentations [54] and consistent improvement has been achieved with these studies. However, it is intractable to eliminate every shortcut and sometimes tricky to craft the correct positive pairs. Different from these methods, we show that augmentations that perturb the semantic features and craft negative samples can be more effective to impose additional regularization. For example, to prevent models from relying on local features, it is much easier to destroy global features and create negatives than to remove all the local features and create positives. Furthermore, by maximizing the difference between natural images and their non-semantic versions in the representation space, the models are coerced to avoid any potential shortcuts shared by them.

Methods like MoCo [18] and SimCLR [7] distinguish positive pairs from negative pairs that are picked from the rest of the dataset. However, most of the negatives prove to be unnecessary and insufficient [13, 31]. To excavate effective negative samples, these methods heavily depend on the large batch sizes [7] or memory bank [18]. Utilizing hard negative samples has long been recognized as an effective approach to boost model performance [17, 29, 55, 46]. In the contrastive learning studies, [10, 43] modify the contrastive learning loss to make it assign greater weights to the hard negative samples. [31] proposes to synthesize hard negative samples by taking linear combinations of the hardest negative samples. Our work is orthogonal to these ideas in the way that we propose to generate negative samples from given images themselves to reduce the reliance on the undesired features. In addition, two recent works [25, 32] study the application of adversarial examples as hard positive and negative samples in contrastive learning. [45] augments the images by manipulating their foregrounds and backgrounds to generate negative and positive samples. Compared with these studies, we mainly focus on the OOD evaluation of the models. In addition, our patch-based augmentation is also related to the self-Supervised learning methods that adopt the pretext task based on jigsaw [41, 39, 20], which we discuss in the Appendix B.2.

**Robustness and out-of-domain generalization of CNNs** High test accuracy provides no guarantee that a network learns high-level semantic features instead of low-level superfluous features that exist in both training and test dataset [30]. An increasing number of studies have corroborated such concerns and found that CNNs can rely on local patches [4, 3], texture [14], high-frequency components [51] and even artificially added features [26] to achieve high test accuracy. These superficial correlations become brittle under large domain shifts [22, 21]. This still remains an unsolved problem [47] and is rarely discussed in the contrastive learning setting.

Among all these undesired features, the shape-texture bias has been widely discussed in recent studies [14, 36]. Previous work has shown that CNNs trained on the ImageNet dataset are biased to texture features and such over-reliance can hurt the generalization performance of CNNs [14, 24]. Several studies have aimed at mitigating this problem [38, 35] or providing a better understanding [23, 27]. In this paper, we introduce a dataset called ImageNet-Texture, which can help future studies on these problems. Our method also effectively controls the trade-off between shape and texture bias. We provide new insights about this problem based on the analysis of our method and dataset.

## 6 Closing Remarks

**Conclusion** CNNs are prone to learn discriminative features that are vulnerable under domain shifts. In this paper, we first demonstrate the regularization power of contrastive learning to discard any undesired features by generating appropriate negative samples. We explore two approaches, the patch-based and texture-based augmentations, to craft negative samples with only local features preserved. We show that the representations learned by contrastive learning with such negative samples depend less on the local features, and consequently generalize better under OOD settings. We hope this paper can encourage people to rethink the role that negative samples play in contrastive learning, which hopefully leads to more efficient methods to generate negative samples.

**Limitations** The problem of dependence on superficial features exists in various domains beyond vision, such as language [37, 28]. Therefore, it is intriguing to consider generalizing such an idea to other modalities. In addition, as the mechanism to ensure that contrastive learning models trained on large datasets to discard the bias of the datasets is yet to be invented, severe social issues in fairness or privacy may be raised as a result [5, 33]. In this paper, we show how to calibrate the bias towards texture features using proposed negative samples. It is also worth considering whether contrastive learning can be used to address any of these negative effects triggered by bias in datasets.

## Acknowledgments and Disclosure of Funding

The authors thank the National Science Foundation, grant no. IIS-1910132 and the Guaranteeing AI Robustness Against Deception (GARD) program from DARPA for their support of this project. The authors thank Yannis Kalantidis for his help with reproducing MoCo-v2 on the ImageNet-100 dataset and Joshua Robinson for providing the checkpoints of IFM models for comparison.

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
