# A ImageNet-Texture

See Figures 7 and 8 for examples of the ImageNet-Texture dataset and their counterparts in the original ImageNet dataset. The analysis of models pretrained on ImageNet and evaluated on ImageNet-Texture is presented in Section A.2.

## A.1 Examples

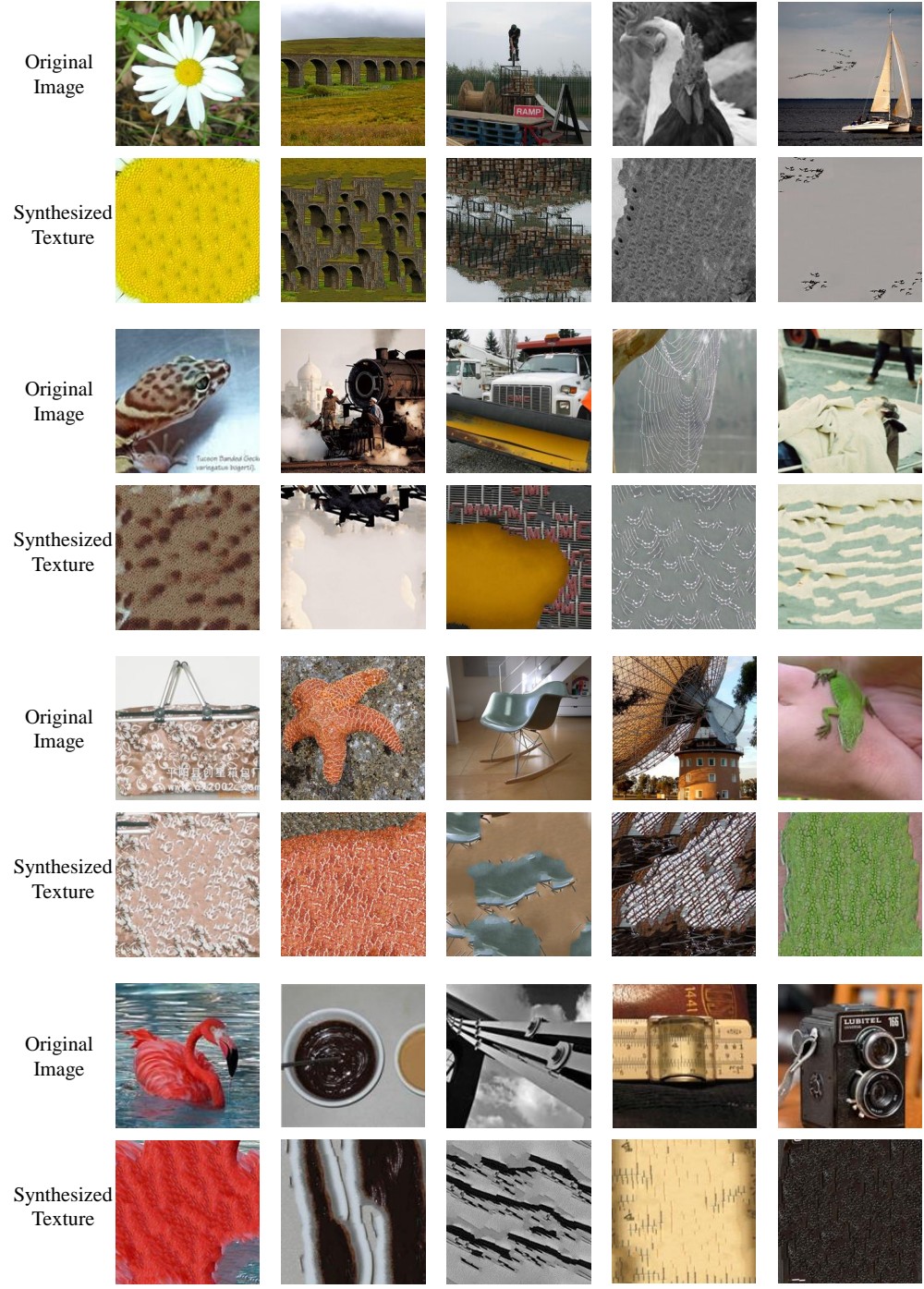

Figure 7: Randomly picked images from our ImageNet-Texture dataset and their corresponding original images from the ImageNet dataset.

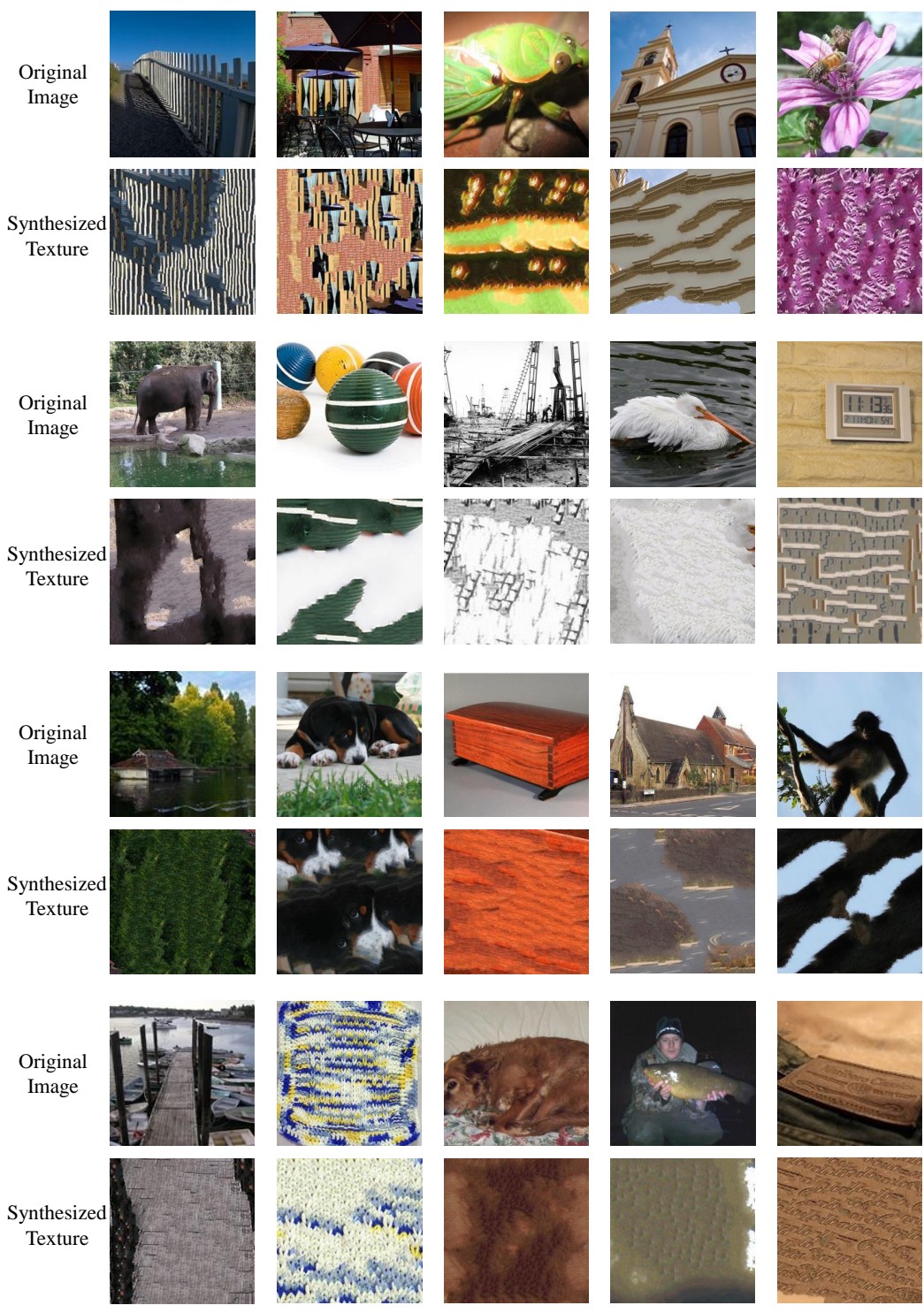

Figure 8: More randomly picked images from our ImageNet-Texture dataset and their corresponding original images from the ImageNet dataset.

## A.2 Analysis

The proposed ImageNet-Texture dataset contains images with preserved texture information and diminished shape information, which can be used to better understand how models behave when only texture features are available. We evaluate the pretrained models provided in Pytorch model zoo [2] on the ImageNet-Texture versus standard ImageNet and report the accuracy in Figure 9a. As shown in the figure, VGG networks generally have a higher accuracy on the ImageNet-Texture dataset. For the rest of the model architectures, we see a positive correlation between the standard accuracy and texture accuracy.

We also plot the histogram of the classes with different accuracy based on a pretrained ResNet-50 model in Figure 9b, which achieves a relatively high accuracy, $9.3\%$, among all the models. The class-based accuracy denotes the accuracy on the images of the same certain class. We find that on the 343 out of 100 classes, the model can only achieve an accuracy smaller than $2.5\%$, which shows that texture feature only is not sufficient to classify these classes. But we do notice a long tail of the distribution. Specifically, 15 classes have an accuracy larger than $50\%$. Sample images of these classes are demonstrated in Figure 10. Notably, texture plays an important role in distinguishing these classes. Shape is often less well-defined in these classes, for example in window screen and rapeseed.

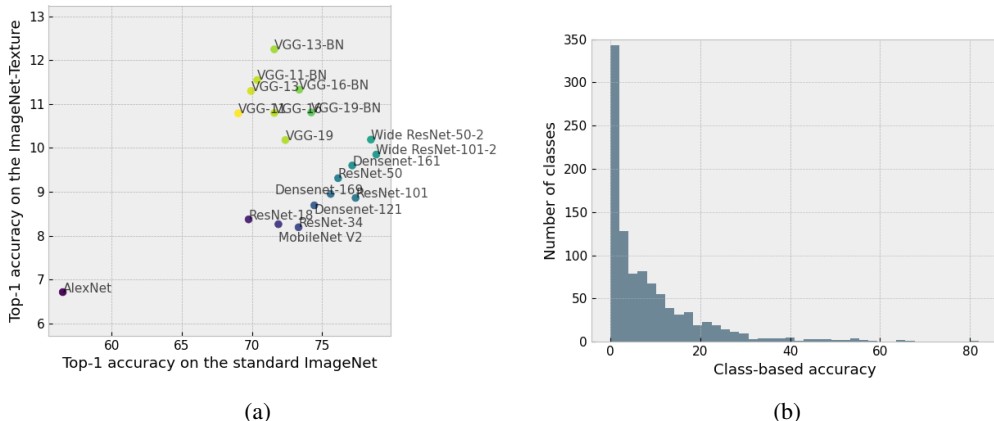

(a)                      (b)

Figure 9: (a) Accuracy on the ImageNet-Texture versus standard ImageNet with different model architectures. (b) Histogram of accuracy on the individual ImageNet-Texture classes with ResNet-50.

## B    Discussion on the contrastive learning with non-semantic negatives

### B.1    Comparison of two ways to apply $\alpha$ in NCE loss

Since the non-semantic negative samples play different roles from the original negative samples, we introduce an additional parameter to control the penalty w.r.t. non-semantic negatives. There are two straightforward alternatives to implement $\alpha$, which we call $\mathcal{L}_{in}$ and $\mathcal{L}_{out}$:

$$\mathcal{L}_{in} = -\sum_{i \in I} \log \frac{\exp\left(z_i \cdot z_p / \tau\right)}{\exp\left(z_i \cdot z_p / \tau\right) + \exp\left(\alpha z_i \cdot z_{ns} / \tau\right) + \sum_{n \in \mathcal{N}} \exp\left(z_i \cdot z_n / \tau\right)}$$

$$\mathcal{L}_{out} = -\sum_{i \in I} \log \frac{\exp\left(z_i \cdot z_p / \tau\right)}{\exp\left(z_i \cdot z_p / \tau\right) + \alpha \exp\left(z_i \cdot z_{ns} / \tau\right) + \sum_{n \in \mathcal{N}} \exp\left(z_i \cdot z_n / \tau\right)}$$

We compare the relative importance of different pairs play in the gradient w.r.t. $z_i$:

$$\frac{\partial \mathcal{L}_{in}}{\partial z_i} = \frac{z_p / \tau \exp\left(z_i \cdot z_p / \tau\right) + \alpha z_{ns} / \tau \exp\left(\alpha z_i \cdot z_{ns} / \tau\right) + \sum_{n \in \mathcal{N}} z_n / \tau \exp\left(z_i \cdot z_n / \tau\right)}{\exp\left(z_i \cdot z_p / \tau\right) + \exp\left(\alpha z_i \cdot z_{ns} / \tau\right) + \sum_{n \in \mathcal{N}} \exp\left(z_i \cdot z_n / \tau\right)} - z_p / \tau$$

---

[2] https://pytorch.org/vision/stable/models.html

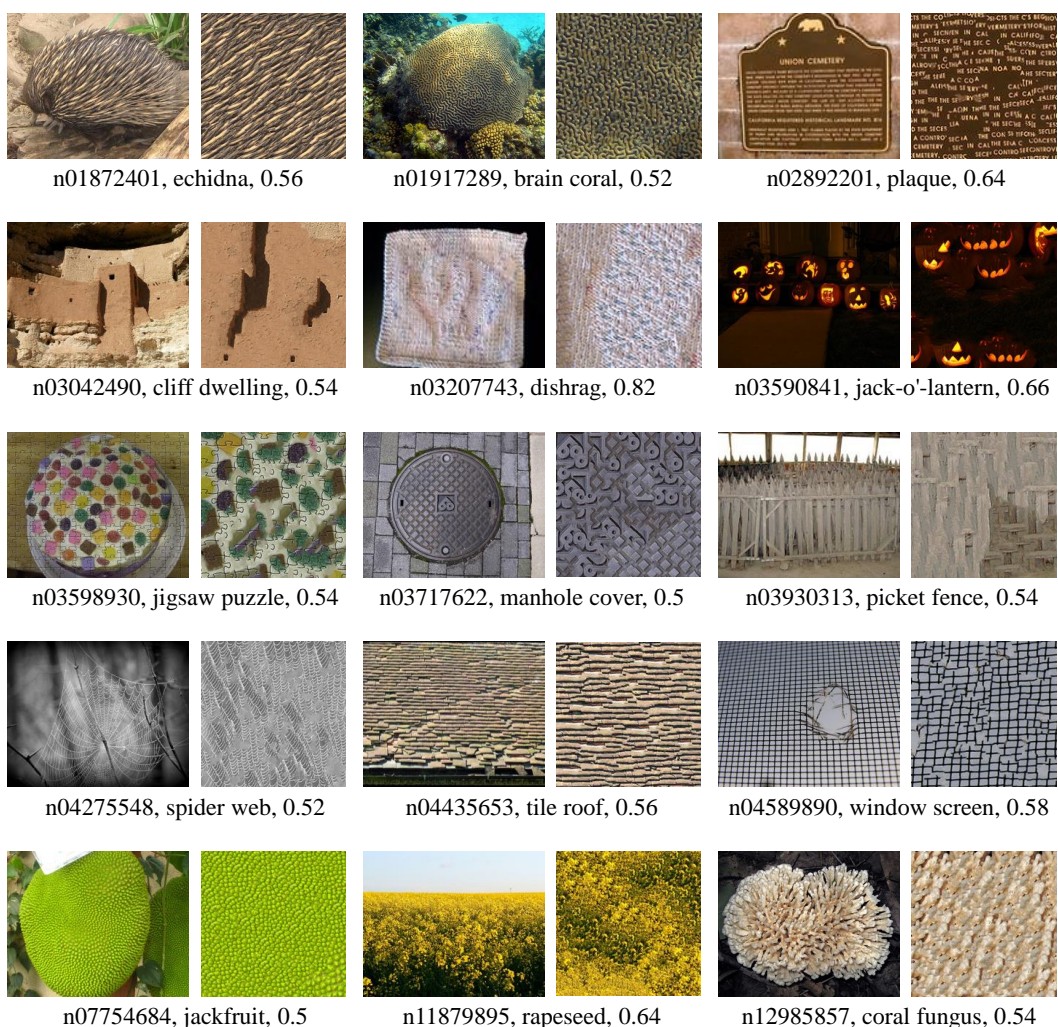

Figure 10: On some classes, a ResNet-50 model trained on the standard ImageNet dataset can achieve $> 50\%$ accuracy when only texture features are available. For each class, one sample image and its texture version are displayed. The caption of each image pair indicates the ImageNet class ID, class name and ResNet-50 accuracy on the texture images.

$$\frac{\partial \mathcal{L}_{out}}{\partial z_i} = \frac{z_p/\tau \exp\left(z_i \cdot z_p/\tau\right) + \alpha z_{ns}/\tau \exp\left(z_i \cdot z_{ns}/\tau\right) + \sum_{n \in \mathcal{N}} z_n/\tau \exp\left(z_i \cdot z_n/\tau\right)}{\exp\left(z_i \cdot z_p/\tau\right) + \alpha \exp\left(z_i \cdot z_{ns}/\tau\right) + \sum_{n \in \mathcal{N}} \exp\left(z_i \cdot z_n/\tau\right)} - z_p/\tau$$

Since the denominator normalizes the 3 kinds of pairs equally, we only pay attention to the numerator. The difference between $\mathcal{L}_{out}$ and $\mathcal{L}_{in}$ is that $\mathcal{L}_{out}$ has $\alpha$ inside the exponential. Because of the exponential tail, it applies a exponentially larger weight to the negatives that are harder. Since non-semantic negatives are often harder as shown in Figure 2, $\mathcal{L}_{in}$ regularizes the non-semantic negatives more effectively than $\mathcal{L}_{out}$. Note that the implementation of $\mathcal{L}_{out}$ is equivalent to using $\alpha$ non-semantic negatives samples for each input image. In addition, the number of standard negative $|\mathcal{N}|$ in the loss also affects the relative importance of non-semantic. The smaller $|\mathcal{N}|$ is, the larger effect is played by the non-semantic negative.

## B.2 Comparison of patch-based negatives and jigsaw-based pretext tasks

Our patch-based augmentation is also closely related to some of the self-supervised learning methods which solve jigsaw as the pretext task. Specifically, [41] proposes to learn meaningful representations through predicting the order of shuffled 3x3 patches of a given image. [39] further extends this idea

to contrastive learning and learn representations that are invariant under such pre-text transformation. [20] exploit the connection between local patches to learn the representations by fusing it with contrastive predictive coding. The common idea behind these methods is to learn the representations based on the correct configuration of the patches in the natural images. However, our patch-based augmentation treats the entire image with unsorted patches as a wrong configuration of the a natural image and use it as the negative samples in the contrastive learning.

## C Experimental details and additional results

### C.1 Implementation details

For ImageNet-1K dataset, we follow the official hyperparameters to MoCo-v2 [8]. For ImageNet-100 and STL-10 datasets, we follow [31] which uses a different memory bank size of 16384 and batch size 128 during the pretraining. For linear evaluation, we adopt a learning rate 10.0, batch size 128, and a learning schedule that decreases the learning rate by 0.1 at 30, 40, and 50 epochs. For BYOL pretraining, we follow [16] to utilize the LARS optimizer with cosine learning rate schedule with a global weight decay parameter of $10^{-6}$ and momentum of 0.9. We use a batch size of 1024 and an initial learning rate 4.8. For both MoCo and BYOL, the non-semantic negatives are input to the momentum branch. All of our models are trained on 4 GTX 1080 Ti gpus. Training single model takes around 1.5 and 12 days on the ImageNet-100 and ImageNet-1K datasets respectively.

### C.2 Results on ImageNet-C with different levels of severity

We show the accuracy of different MoCo-v2 models on the ImageNet-C-100 dataset with different corruption severity levels in Table 5. Specifically, as we have seen in the main body of the paper, a larger $\alpha$ often favors larger domain shift. This is further demonstrated by the fact that the model with $\alpha = 3$ outperforms the model with $\alpha = 2$ on the highest corruption level 5.

| Severity level | 1 | 2 | 3 | 4 | 5 |
|---|---|---|---|---|---|
| MoCo-v2 - $k = 16384$ | $65.58_{\pm 0.32}$ | $53.41_{\pm 0.27}$ | $42.31_{\pm 0.31}$ | $31.87_{\pm 0.31}$ | $22.22_{\pm 0.20}$ |
| + Texture-base NS | $66.00_{\pm 0.47}$ | $54.11_{\pm 0.43}$ | $42.96_{\pm 0.41}$ | $32.23_{\pm 0.26}$ | $22.60_{\pm 0.23}$ |
| + Patch-base NS - $\alpha = 2$ | $\mathbf{67.84}_{\pm 0.21}$ | $\mathbf{55.99}_{\pm 0.20}$ | $\mathbf{44.65}_{\pm 0.40}$ | $\mathbf{33.74}_{\pm 0.51}$ | $23.41_{\pm 0.50}$ |
| + Patch-base NS - $\alpha = 3$ | $65.82_{\pm 0.04}$ | $54.95_{\pm 0.13}$ | $44.10_{\pm 0.17}$ | $33.67_{\pm 0.25}$ | $\mathbf{23.71}_{\pm 0.28}$ |
| MoCo-v2 - $k = 8192$ | $65.44_{\pm 0.52}$ | $53.52_{\pm 0.49}$ | $42.71_{\pm 0.42}$ | $32.09_{\pm 0.33}$ | $22.36_{\pm 0.19}$ |
| + Patch-base NS - $\alpha = 2$ | $\mathbf{68.01}_{\pm 0.01}$ | $\mathbf{56.26}_{\pm 0.12}$ | $\mathbf{45.04}_{\pm 0.28}$ | $\mathbf{34.16}_{\pm 0.34}$ | $\mathbf{23.92}_{\pm 0.27}$ |

Table 5: Top-1 accuracy on ImageNet-C-100 dataset with different levels of severity.

### C.3 Additional ablations on the patch-based augmentations

#### C.3.1 Patch sizes

We show more results on the patch-based augmentations with different patch sizes by reporting the test accuracy on the ImageNet-100 validation set and corresponding ImageNet-Sketch dataset in Figure 11. Specifically, we sample patch sizes from uniform distributions $d \sim \mathcal{U}(x, y)$ of different interval $[x, y]$. We find that for the performance on the standard validation set, both the large or smaller patch sizes cause a less desired accuracy. But for the ImageNet-Sketch dataset, the larger patch sizes generally provide larger performance improvement. Our conjecture on this observation is that using larger patch sizes prevents the model to learn some of the local features that are shared between training and validation set while absent from the sketch dataset.

#### C.3.2 Data augmentations on individual patches

We test whether common data augmentation methods can be used to augment individual patches so as to further improve the performance. We report the accuracy on the ImageNet-100 validation set with the model trained with the patch-based negative samples in Table 6. The patches are potentially augmented with horizontal flip or vertical flip with 50% probability, or rotation in $0°, 90°, 180°$ or $270°$ with 25% probability respectively. We find that for a larger patch size, i.e. $d \sim \mathcal{U}(16, 72)$, such

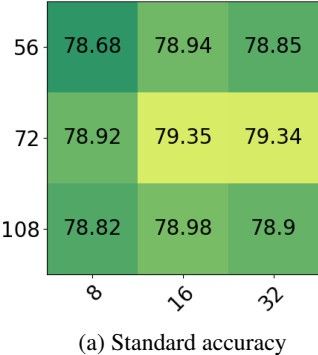

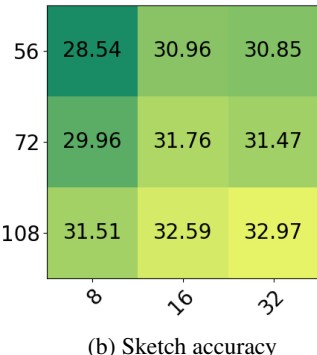

| (a) Standard accuracy | (b) Sketch accuracy |

Figure 11: Test accuracy of patch-based non-semantic augmentations with patch sizes sampled from different uniform distribution. The x-axis indicates the lower boundary of the sampling interval and the y-axis indicates the higher boundary.

augmentations always downplay the accuracy. But for a smaller patch size, such augmentations, especially with horizontal flip only, can improve the performance. For the larger patch size, we conjecture that it is important to ensure that the patches to be identical to the certain part of the input image. Therefore the non-semantic features are best preserved. However, for a smaller patch size, such augmentations have a less impact on the captured non-semantic information.

| Patch size | No aug. | + Horizontal flip | + Vertical flip | + Rotation |
|---|---|---|---|---|
| $8 - 28$ | 78.58 | 78.72 | 78.66 | 78.64 |
| $16 - 72$ | 79.35 | 79.06 | 78.92 | 78.40 |

Table 6: Accuracy on ImageNet-100 validation set with different configurations of augmentations on the individual patches. Augmentations are cumulative across the columns (e.g. the "+ Vertical flip" model used horizontal and vertical flip).

### C.4 BYOL with different kinds of negative samples

The proper way to add negative samples to BYOL is still an open research problem. In this section, we provide additional experiments to demystify the role of non-semantic negative samples played in BYOL. Specifically, we train BYOL with a regular negative that is randomly picked from the same batch with the query sample excluded, or a regular negative as well as a patch-based non-semantic negative. We report the accuracy on the ImageNet-100 variants in the Table 7. We find that introducing a regular negative sample without further modification dramatically undermines the performance, which demonstrates that the improved performance is due to the negative mining strategy instead of adding an arbitrary negative sample.

| | ImageNet | ImageNet-C | ImageNet-S | Stylized-ImageNet | ImageNet-R |
|---|---|---|---|---|---|
| BYOL | 78.76 | 44.43 | 35.84 | 15.01 | 39.53 |
| + Patch-based | 78.81 | 44.60 | 36.76 | 15.52 | 41.16 |
| + Reg. | 57.22 | 24.19 | 19.11 | 6.74 | 20.47 |
| + Patch-based and Reg. | 57.24 | 24.37 | 20.11 | 6.60 | 20.95 |

Table 7: Top-1 accuracy of BYOL trained with different negatives on the ImageNet-100 datasets.

### C.5 Hardness of texture-based negative samples

We show the distribution of similarities between the representations of input image versus different paired samples in Figure 12. The similarities are calculated by the models trained without (blue) and with (red) texture-based negative samples. We use $\alpha = 2$ when texture-based negative samples are

adopted. Comparing with Figure 3, the regularization of texture-based negative samples is generally weaker than the patch-based negative samples. For example, the average similarity w.r.t patch-based negative samples decrease from $0.4040$ to $0.3677$ when using texture-based negative samples for training, which is $0.3252$ when using patch-based negative samples.

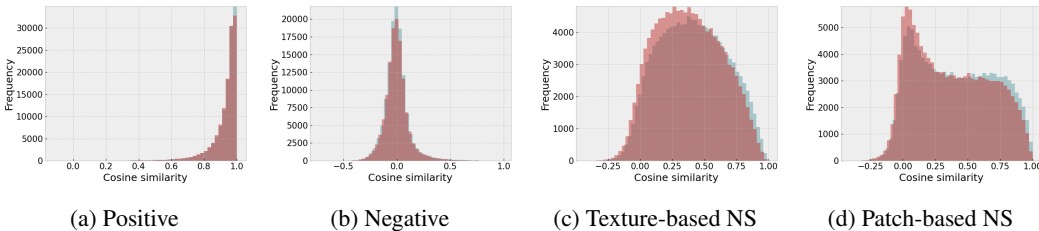

| (a) Positive | (b) Negative | (c) Texture-based NS | (d) Patch-based NS |

Figure 12: The histogram of cosine similarity between the representations of different kinds of pairs using models trained without (blue) and with (red) texture-based negative samples.

### C.6 Hardness of patch-based negative samples using models trained with different $\alpha$

We show the distribution of similarities calculated by the models trained with patch-based negative samples and different $\alpha$ in Figure 13. When $\alpha$ increases from 0 to 5, we see a larger penalty on the similarities to both patch-based and texture-based negative samples. When $\alpha = 5$, the distributions of two negative samples are close to a normal distribution like the distribution of the standard negative samples. Specifically, the average similarity w.r.t. the patch-based negative samples further decreases to $0.4040$ to $0.1184$. In terms of the original pretext task, we find that the distribution of both standard negative and positive samples become wider and less concentrated to $0$ and $1$ respectively. This shows that a larger $\alpha$ makes the optimization of the original objective more difficult.

### C.7 More results on ImageNet-1K

|  | ImageNet | ImageNet-C | ImageNet-S | Stylized | ImageNet-R |
|---|---|---|---|---|---|
| MoCo-v2 [8] | 67.60 | 87.7 | 17.47 | 5.55 | 27.81 |
| + MoCHi (128,1024,512) [31] | 66.62 | 90.0 | 16.00 | 5.32 | 25.29 |
| + MoCHi (512,1024,512) [31] | 67.56 | 88.7 | 16.32 | 5.94 | 25.71 |
| + Patch-base NS - $\alpha = 2, k = 65536$ | **67.92** | 87.6 | 18.58 | 6.34 | 28.95 |
| + Patch-base NS - $\alpha = 3, k = 65536$ | 60.80 | 92.1 | 18.79 | **6.80** | 28.11 |
| + Patch-base NS - $\alpha = 2, k = 32768$ | 67.83 | **87.2** | 18.70 | 6.06 | 28.50 |
| + Patch-base NS - $\alpha = 2, k = 16384$ | 67.34 | 87.5 | **19.49** | 6.49 | **29.45** |

Table 8: Top-1 accuracy on the ImageNet-1K dataset and its sketch, stylized, rendition variants, and mCE on the ImageNet-C dataset.

We show more results on the ImageNet-1K dataset using MoCo-v2 models trained with patch-based negative samples in Table 8. We also report the other released MoCHi model [3], and the different hyperparameters are indicated following the model name in the table. For our method, we focus on the models when a larger penalty on the similarity w.r.t. the patch-based negative samples is added. Specifically, we report the results with a larger $\alpha = 3$ and smaller memory bank sizes $k = 32768$ or $k = 16384$. As shown in the table, increasing the penalty stably improves the generalization under OOD settings. For example, when $\alpha = 2$ and $k = 16384$, the accuracy on ImageNet-S and ImageNet-R increase from $17.47$ to $19.49$ and $27.81$ to $29.45$ respectively.

---

[3] https://europe.naverlabs.com/research/computer-vision/mochi/

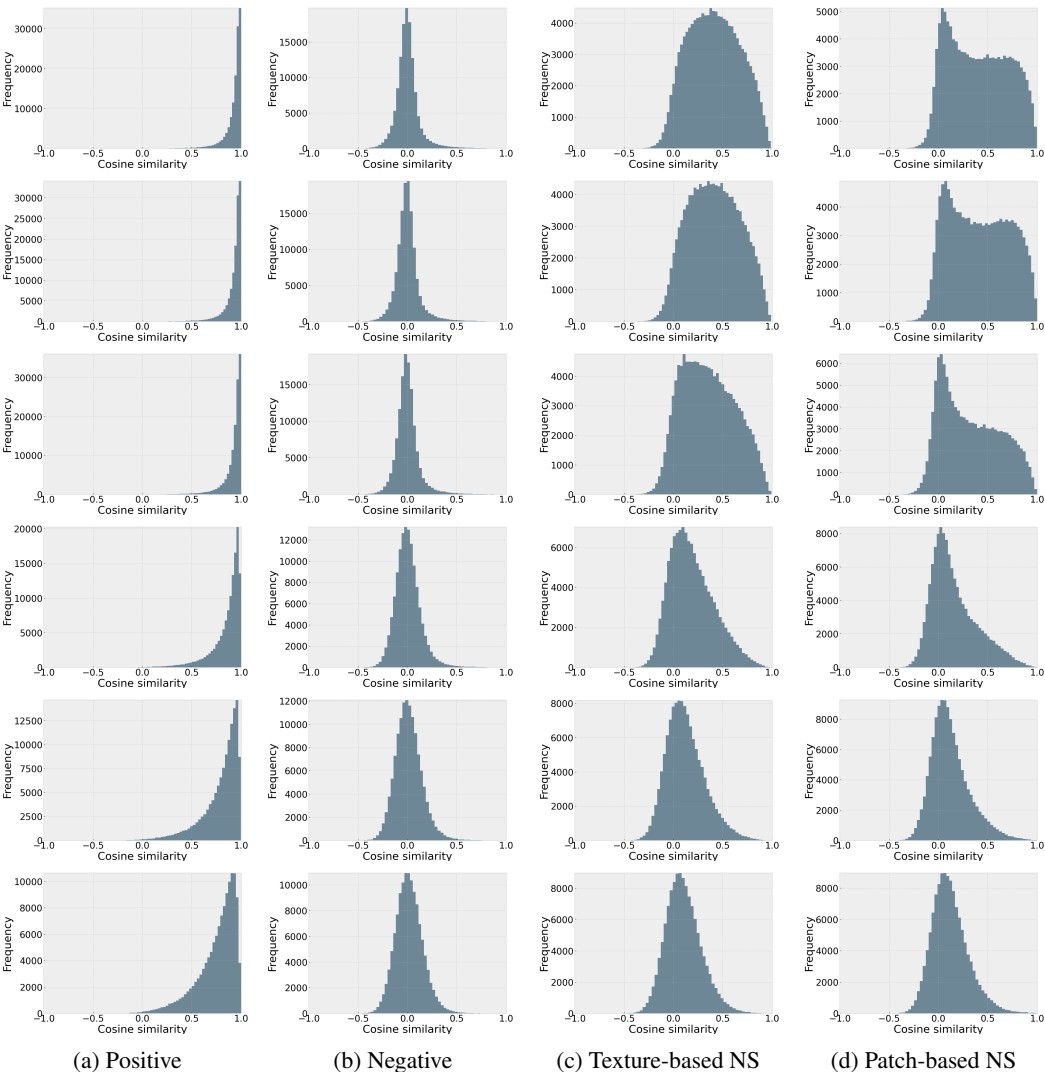

(a) Positive      (b) Negative      (c) Texture-based NS      (d) Patch-based NS

Figure 13: The histogram of cosine similarity between the representations of different kinds of pairs using models trained with patch-based negative samples and different $\alpha$ values from $0$ (top) to $5$ (bottom).