# OpenReview forum: "Robust Contrastive Learning Using Negative Samples with Diminished Semantics"
_NeurIPS.cc/2021/Conference — NeurIPS 2021 Poster_

### Official Review · Reviewer_zVco · 2021-07-15

**Rating:** 6
**Confidence:** 4

**Summary:**

The paper proposes to augment current contrastive self-supervised learning (SSL) methods such as BYOL and MoCo with additional negative samples that are augmented so that they remove semantics from the original image, thus forcing the learning of more robust representations that focus less on superficial image features. The authors propose to do so by generating negative samples based on textures and patches from an original image. The experiments suggest that models learned with the augmented negative samples generalize better.

**Ethical Concerns:**

I do not see direct ethical concerns with the paper and the authors briefly mention concerns in the paper.

**Limitations And Societal Impact:**

The authors briefly address the limitations and potential negative societal impact of their work.

**Main Review:**

**Originality**

The method proposes to augment the negative samples of constrative SSL methods. This has been done before in the literature ([31], [43] for example), but not necessarily for textures or image patches. The role of texture and patch-based features for supervised learning models has been studied before (e.g. [3]). Therefore, it is a combination of ideas that have been investigated before, but in its current form it is relatively novel.

**Clarity**

The paper is well written. The claims and hypothesis of the authors are clearly stated, and the experiments and analysis is easy to follow.

**Quality**

The premise of the paper is clear, and the experimental section is quite extensive. That being said, there are some issues with the approach.

First, the modification of the BYOL objective is ad-hoc and not really justified. An ablation that also shows the difference in using a regular negative sample or a texture-based negative sample for this method would be useful to understand whether the reported improved performance is due to adding a negative sample or because of the particular mining strategy for the negative. Additionally, the improved performance for the augmented model on a few datasets (ImageNet-100 e.g.) is quite below the standard deviation of the runs and therefore not really significant.

Second, for most approaches the introduction of another negative sample effectively increases the batch size of the training model or equivalently, the model is trained on more samples. An ablation showing that this does not have an effect on the regular approach would be useful.

Thirdly, and following the previous point, some of the reported metrics of the baseline methods do not seem to match the published results and seem to be slightly lower. For example, MoCoV2 reports improved performance well above 67.60 on Imagenet 1K when trained for longer, and MoCHi reports better performance over the MoCoV2 baseline and in this paper its reported worse performance. It would be useful to have experiments with the same setup/performance as in the original papers, and with configurations that maximize the performance of a model to make sure that the reported improvements are significant and not necessarily due to differing training setups such as feeding more examples to the model.

**Significance**
For the above reasons, it is not fully well supported that the proposed method extensions and negative mining strategies show a significant advantage to the baseline methods (considering also the more complex setup involved), however, even with these doubts the paper might be of interest to the community and therefore my favorable rating.

**Review updates**
After reading the rest of the reviews and the authors' response, I am still not convinced about the results using a modified BYOL and the performance discrepancy with MoCHI. However, as stated in the original review, I believe the paper might be of interest to the community despite these issues and therefore stand by my original rating.

**Time Spent Reviewing:**

3

---

> ### Author Response · Authors · 2021-08-10
> **Response to Reviewer zVco**
>
> Thank you for your comments and time!
>
> ---
> **“First, the modification of the BYOL objective is ad-hoc and not really justified. An ablation that also shows the difference in using a regular negative sample or a texture-based negative sample for this method would be useful...”**
>
> We had tried a few different ways to add non-semantic samples to BYOL objective such as training the projector and predictor to minimize the distance of non-semantic negative pairs while adversarially training the encoder to maximize the distance, given that only the encoder will be used for fine-tuning. This is equivalent to optimizing the encoder to maximize a neural distance defined by the projector and predictor, which is generally harder than directly optimizing the L2 distance. But we didn't see any improvements brought by this over the simple idea. In general, we think there is an open question on what the proper way to add negative samples to BYOL is. Given its popularity, we provide a simple way as a preliminary experiment, although it is less significant than other results on MoCo-v2.
>
> We would like to thank the reviewer for the suggestion on the ablation. We followed the idea to run two experiments to further demystify this problem: we trained BYOL using a regular negative (picked from the same batch), or a regular negative as well as a patch-based non-semantic negative. As shown in the Table below, introducing a regular negative sample seems to dramatically undermine the performance, which shows that the improved performance is not due to adding a negative sample but the mining strategy.
>
> |                        | ImageNet   | ImageNet-C | ImageNet-S | Stylized-ImageNet | ImageNet-R |
> |------------------------|------------|------------|------------|-------------------|------------|
> | BYOL                   | 78.76 | 44.43 | 35.84 | 15.01        | 39.53 |
> | + Patch-based          | 78.81 | 44.60 | 36.76 | 15.52        | 41.16 |
> | + Reg.                 | 57.22      |     24.19       | 19.11      | 6.74              | 20.47      |
> | + Patch-based and Reg. | 57.24      |    24.37        | 20.11      | 6.60              | 20.95      |
>
> ---
> **“Second, for most approaches the introduction of another negative sample effectively increases the batch size of the training model or equivalently, the model is trained on more samples. An ablation showing that this does not have an effect on the regular approach would be useful.”**
>
> Our method only increases the batch size / memory bank size by 1 while largely exceeding the benefit brought by introducing extra regular negatives, which can be seen in Figure 5 - when memory bank size is 8192, the improvement on validation accuracy brought by adding another **8192** regular negative samples (+0.15) is much smaller than adding **1** non-semantic negative samples (+1.81), and such gap is even larger on the OOD datasets.
>
> ---
> **“... some of the reported metrics of the baseline methods do not seem to match the published results and seem to be slightly lower. For example, MoCoV2 reports improved performance well above 67.60 on Imagenet 1K when trained for longer, and MoCHi reports better performance over the MoCoV2 baseline and in this paper its reported worse performance...”**
>
> Since we didn’t find any reported numbers on the OOD datasets and due to the limited computational resource we have, we used the officially released models to calculate all the numbers (line 223), which seems to give inconsistent numbers with the published ones. We did try to implement MoCHi ourselves but weren't able to reproduce their reported results. We tried to reach out to the authors for the code but we were told that they couldn't release it for commercial reasons. Therefore, we used one of the two models they released that gave better numbers in our main text. If the reviewer has any pointers to reproduce MoCHi, we would like to try and include it in the revision.

---

### Official Review · Reviewer_ffpV · 2021-07-16

**Rating:** 6
**Confidence:** 4

**Summary:**

This paper proposes new augmentation strategies for contrastive learning that produce negative samples designed to mimic the texture of points, but not have their shape data. The goal is to make models not rely on texture data. The authors report extensive experiments on the effectiveness of their method, and discuss how different image classes rely on different features in the shape-texture trade-off.

**Limitations And Societal Impact:**

Yes

**Main Review:**

This is a nice paper with strong central ideas and extensive experiments. The shape-texture trade-off is interesting and important, and I'm glad to see work using contrastive learning to try to control the trade-off.

I think the work could be improved if the authors spend a bit more time explaining the results and discussing where the lift comes from. The experiments are extensive, but there are multiple effects that are not explained. The two major questions that stick out to me are:

1. In table 1, there are some tasks where patch-based with alpha = 2 works better, and some tasks where patch-based with alpha = 3 works better. I could not find any explanation about this discrepancy. *Why* does one method outperform in some cases, but not others? Why does patch-based with alpha = 3 underperform baselines by two points on ImageNet, but alpha = 2 outperforms by two points?

Section 3.1 spends considerable space on exposition of the results, but does not spend enough space discussing how the method creates lift, or under what circumstances to expect lift.

2. The results from Figures 4 and 5 about memory bank size are striking, and I don't think that the text explanation or the analysis in appendix B.1 adequately explain it. Performance consistently increases with memory bank size -- and even appears to grow faster than log-linear in memory bank size -- but then significantly drops at the last tested memory bank size. I would expect a smoother degradation for the explanation that the benefit from larger memory bank size is being outweighed by less impact from the non-semantic negatives (note that increasing memory bank size is still helpful for MoCo). More experiments would be helpful to understand this phenomenon and validate the hypothesis.

**Time Spent Reviewing:**

3

---

> ### Author Response · Authors · 2021-08-10
> **Response to Reviewer ffpV**
>
> Thank you for your comments and time!
>
> ---
> **“In table 1, there are some tasks where patch-based with alpha = 2 works better, and some tasks where patch-based with alpha = 3 works better...Why does patch-based with alpha = 3 underperform baselines by two points on ImageNet, but alpha = 2 outperforms by two points?”**
>
> Thank you for raising this question. On ImageNet, we think that the texture and local features on the ImageNet training set do transfer to its validation set. Therefore, penalizing too much (alpha=3) on the usage of such features undermines the accuracy of it. However, when the information is less transferrable such as the ImageNet-Sketch dataset, the accuracy increases when alpha changes from 2 to 3, i.e. penalizing more brings larger improvement under such OOD scenarios.
>
> We found this to be an interesting research question and to investigate it more, we experimented with a wider range of alpha in {1, 2, 3, 4, 5}. On a dataset with conflicted cues of texture and shape, we found that increasing alpha monotonically decreases the texture accuracy while increasing the shape accuracy, which demonstrates an effective control on the shape-texture bias trade-off of the model. The different behaviors of alpha on the different datasets might be explained by the relative importance of the texture and shape feature in the dataset - texture is less important on the ImageNet-Sketch dataset than the ImageNet validation set. More details can be found in Section 4.
>
> ---
> **“In Figures 4 and 5, performance consistently increases with memory bank size but then significantly drops at the last tested memory bank size... More experiments would be helpful to understand this phenomenon...”**
>
> We thank the reviewer for pointing this out and we agree that more experiments are necessary for a better understanding of this. In general, we think that the log plot may be misleading and make it look like a "sudden drop". Therefore, we ran experiments with evenly distributed points (4096, 6144, 8192, 10240, 12288, 14336, 16384) between 4096 and 16384, and made a plot in a linear scale instead. The preliminary plot can be found here: https://drive.google.com/file/d/1_V8sFmLOvDckpuPYK-L8EqIYHKVWqL65/view?usp=sharing. In general, we found that using patch-based negative samples consistently improves the accuracy of MoCo-v2. The drop is no more clear in the figure. However, we found that though most points fall into the standard deviation interval of the expected curve,  more runs are needed to validate the hypothesis. We plan to keep working on it and include more results in our revision.

---

### Official Review · Reviewer_CTYi · 2021-07-16

**Rating:** 6
**Confidence:** 4

**Summary:**

The paper proposes texture-based and patch-based augmentations to generate negative samples from input images to improve the robustness and generalization ability of contrastive learning. The texture-based augmentation generates negative texture images based on two patches extracted from input images using texture synthesis tools. The and patch-based augmentations constructs non-semantic images by tiling randomly sampled patches of different sizes from the input image. Experiments on ImageNet and its variances demonstrate the effectiveness of the proposed method on improving generalization ability of existing contrastive learning methods.


**Limitations And Societal Impact:**

Yes

**Main Review:**

Originality: The idea of adding negative samples that only share the same texture as the anchor image to avoid learning the texture semantic is novel.

Quality: The paper has the following weaknesses:
1.	The paper lacks comparison with other baselines. For example, in [1], the authors proposed to use texture randomization in the augmentations which can also learn texture-invariant features. Also, in [1] the authors reported higher accuracy of MoCo-v2 on ImageNet-100 (81.0%), while the MoCo-v2 accuracy in this paper is less than 78%.
2.	The paper proposed to learn less about the texture semantics by adding negative samples that only share the texture sematic with the anchor images. However, the texture semantics can still be helpful in some cases. Similar to [1], it would be good to try to ensemble models trained with the proposed negative augmentations and without it and see whether this could improve the performance.
3.	The proposed method is restricted to removing texture semantics and lack of extension to other semantics. One interesting experiment to do is to remove color distortion in the augmentations of positive samples and add negative samples that share the same color distribution of the anchor image to see whether the proposed method could be helpful to avoid the color distribution shortcut.

Clarity: The paper is well written and easy to follow.

Significance: the paper proposes a simple method to construct negative samples to prevent contrastive learning from learning texture information which could affect its generalization ability. However, it is not always desirable to remove the texture semantics, since it is also an important semantics in image classification task. Moreover, the proposed method is restricted to the texture semantics and the authors do not show that it could be extended to other semantics. Therefore, the significance of the paper is limited.


**Time Spent Reviewing:**

2 hours

---

> ### Author Response · Authors · 2021-08-10
> **Response to Reviewer CTYi**
>
> Thank you for your questions and constructive suggestions.
>
> ---
> **“The proposed method is restricted to removing texture semantics and lack of extension to other semantics. One interesting experiment to do is to remove color distortion in the augmentations of positive samples and add negative samples that share the same color distribution of the anchor image to see whether the proposed method could be helpful to avoid the color distribution shortcut.”**
>
> Thanks for raising this issue. On adding negatives sharing the same color distribution, the expected color distribution of our patch based non-semantic samples is identical to that of the anchor images, and the actual distribution of samples will be really close, which is doing exactly what the reviewer suggested. Therefore we ran the experiment of training MoCo-v2 w/ and w/o patch-based non-semantic negatives when removing the color jittering augmentation. We found that after removing the color jitter augmentation, the top1 accuracy MoCo-v2 drops from 77.88 to ***70.44*** while the accuracy of MoCo-v2 w/ patch-based NS only drops from 79.35 to ***76.42***, which shows a significant effectiveness of our patch-based negative samples in preventing the models from learning such color distribution shortcut.
>
> ---
> **“...the proposed method is restricted to the texture semantics and the authors do not show that it could be extended to other semantics.”**
>
> We not only worked on texture features in our paper but also local semantics as texture [2] and local features [3] are two of the most debated features in the literature. In addition, we also show that our patch-based method can generalize to the color distribution as mentioned above.  We agree that the two kinds of features may not represent all of the superficial features. Studying all possible superficial features is beyond the scope of a conference paper, and we leave it for future work.
>
> ---
> **“The paper lacks comparison with other baselines. For example, in [1], the authors proposed to use texture randomization in the augmentations which can also learn texture-invariant features. Also, in [1] the authors reported higher accuracy of MoCo-v2 on ImageNet-100 (81.0%), while the MoCo-v2 accuracy in this paper is less than 78%.”**
>
> The reference [1] was not provided in the review but we believe we have identified the paper: https://arxiv.org/pdf/2008.05659.pdf. The authors studied how to automatically adapt CL models with different invariance assumptions to different tasks such as coarse-grained and fine-grained classification tasks using ***different data augmentations on positive pairs,*** while we focus on designing augmentations by ***mining hard negative pairs under out-of-distribution settings.*** Our main contribution is also not another state-of-the-art contrastive learning algorithm, but a different view to leverage contrastive learning to mitigate the undesired bias in CNNs with thorough experiments to demonstrate this (R-PE6h, R-ffpV, R-zVco). This allows us to inject more inductive-bias that cannot be done using positive augmentations, e.g. to prevent models from learning local features, it is much easier to destroy global features and create negatives than to remove all the local features and create positives. One might say using sketch as positive samples can achieve that, but generating aligned sketches is not trivial itself and introduces new artifacts that are not from the training data.
>
> The 81.0% accuracy reported in [1] trains the model for ***500*** epochs. Due to computational resource limits, we carefully followed the ***200-epoch*** training protocol proposed in [4] that is highly tuned and yields better accuracy than the official hyperparameters, which we believe is still a fair comparison. In addition, the code of the referred paper is not available and we are not able to compare it with the same amount of training resources.
>
> ---
> **“it is not always desirable to remove the texture semantics, since it is also an important semantics in image classification task. Moreover,”**
>
> We agree that texture features are important and this is what we advocated for in the paper (line 290-291): we carefully analyzed when texture features are more important than the other settings, which we think offers insights on how to trade-off between shape- and texture-biased in practice. For example, similar to [1], we confirm that texture is more important for the finer-class classification in Section 4.2. In addition, our analysis of texture features on different out-of-domain datasets in Section 4.1 complements [1]'s findings. As said by R-zVco, we are the first to discuss and leverage hard negatives to improve the OOD robustness of CNNs, which, as said by R-PE6h, could “shed light on the texture-shape bias of existing self-supervised methods”.
>
> ---
> **“The paper proposed to learn less about the texture semantics by adding negative samples that only share the texture sematic with the anchor images. However, the texture semantics can still be helpful in some cases. Similar to [1], it would be good to try to ensemble models trained with the proposed negative augmentations and without it and see whether this could improve the performance.”**
>
> We agree that texture semantics are indispensable in many cases. Similar to many previous works [3] on this problem, we find that properly rectifying the over-reliance on the texture features is useful for model generalization. The ensemble method in [1] seems to be an interesting idea of how to dynamically determine which feature to be valued more in different settings. But again, their code is not released which makes it difficult to reproduce their results in a short time.
>
> [1] Xiao, Tete, et al. "What Should Not Be Contrastive in Contrastive Learning." International Conference on Learning Representations. 2021.
>
> [2] W. Brendel and M. Bethge. Approximating CNNs with bag-of-local-features models works surprisingly well on imagenet. In International Conference on Learning Representations, 2019.
>
> [3] R. Geirhos, P. Rubisch, C. Michaelis, M. Bethge, F. A. Wichmann, and W. Brendel. Imagenet-trained CNNs are biased towards texture; increasing shape bias improves accuracy and robustness. In International Conference on Learning Representations, 2019.
>
> [4] Y. Kalantidis, M. B. Sariyildiz, N. Pion, P. Weinzaepfel, and D. Larlus. Hard negative mixing for contrastive learning. In Neural Information Processing Systems (NeurIPS), 2020.

---

> > ### Comment · Reviewer_CTYi · 2021-08-21
> > **Raise score to 6**
> >
> > The rebuttal resolves my concerns. Therefore, I decide to raise the score to 6. I would suggest the authors to include the experiments in the rebuttal in the paper to make it stronger and more generalized. I would also strongly recommend the author to train the baselines and their proposed methods for 500 epochs on ImageNet-100 since it improves the performance a lot and could serve as a better benchmark for future research.

---

### Official Review · Reviewer_PE6h · 2021-07-17

**Rating:** 7
**Confidence:** 3

**Summary:**

This paper introduces non-semantic negative samples into self-supervised learning. These non-semantic samples force the network to reduce the reliance on texture-based features and focus on semantic ones, which are more general in different scenarios.  Experiments on several image classification datasets verify the robustness of the proposed method. In addition, texture-shape bias trade-off is also thoroughly analyzed.

**Limitations And Societal Impact:**

1. As shown in Table 1 and discussed in Section 4, models are sensitive to the penalty hyperparameter $\alpha$ and different datasets may require different settings. However, tuning a pre-trained model for each scenario is not affordable.  And using the same setting for all scenarios may lead to sub-optimal results or even hurt the performance. Discussions about how to apply the proposed methods in real scenarios should be added.


2. Using the non-semantic samples with MoCo or BYOL require extra computation cost for these samples. Time and GPU memory costs should be discussed in the paper.

**Main Review:**

This paper aims to eliminate CNNs’ reliance on low-level features and improve the generalization ability of models. To achieve this, two negative sample generation methods are proposed. These methods construct negative samples which only preserve non-semantic features. This idea is straightforward and sound.

Experiments on multiple datasets show the robustness of models trained with non-semantic samples, verifying the efficacy of the proposed methods. Additionally, detailed analysis on the texture-shape bias trade-off is provided.

In general, this paper is well written and it sheds light on the texture-shape bias of existing self-supervised methods. The improved generalization performance would help self-supervised pre-trained models to be applied in real scenarios.


**Time Spent Reviewing:**

6

---

> ### Author Response · Authors · 2021-08-10
> **Response to Reviewer PE6h**
>
> Thank you for your comments and time!
>
> ---
> **“...models are sensitive to the penalty hyperparameter and different datasets may require different settings... Discussions about how to apply the proposed methods in real scenarios should be added.”**
>
> We totally agree that the optimal values of the hyperparameters are often dataset-dependent. In Table 1, although different choices of alpha may provide the best performance on different datasets, we can choose a single value of alpha and achieve improved performance on all the datasets. For example, choosing alpha = 2 for MoCo-v2 improves performance on all of them. In our study, there is a wide range of values like this.  Therefore, we believe it is still an acceptable option for practical use.
>
> To seek the optimal value, our analysis in Section 4 would provide some heuristics: larger alphas help more in the scenarios where the texture feature is less important such as the ImageNet-Sketch and Stylized-ImageNet datasets, as well as coarse-class classification tasks. We think determining how to predict the importance of texture features under different scenarios is an open and interesting research question, which could be worth more investigation.
>
> ---
> **“Using the non-semantic samples with MoCo or BYOL require extra computation cost for these samples. Time and GPU memory costs should be discussed in the paper.”**
>
> We discussed the GPU usage in Section C.1. In general, since we only use 1 extra non-semantic sample per input sample, the extra memory and computational cost introduced by our method is negligible.

---

### Author Response · Authors · 2021-08-10
**General Response**

We would like to thank all the reviewers for their thoughtful comments. We are encouraged to see that reviewers generally appreciated the strengths of our paper -- the importance of studying OOD robustness and shape-texture trade-off in SSL (R-PE6h, R-ffpV, R-zVco), the straightforwardness (R-PE6h), novelty (R-CTYi, R-ffpV, R-zVco), and soundness (R-PE6h, R-ffpV) of the proposed idea, the extensiveness of our experiments and analysis (R-PE6h R-ffpV, R-zVco) and writing quality (R-PE6h, R-CTYi, R-zVco). Below, we address each reviewer’s comments.

---

### Decision · Program_Chairs · 2021-09-27

**Decision:**

Accept (Poster)

**Comment:**

This paper copes with generating negative samples for contrastive learning, separating the signal from texture and shapes. The idea has been judged sufficiently novel and sound. The experiments are convincing, and the additional results presented in the rebuttal improved the empirical validation further. I recommend this paper for acceptance, and recommend the authors take into account all reviewer’s comments for preparing the final version of the manuscript.